# Territorial Distribution of EU Funds Allocation for Developments of Rural Romania during 2014–2020

**Doru Maier [1],\*, Ancuta-Nicoleta Remete [1], Alina-Mihaela Corda [1], Ioana-Alexandra Nastasoiu [1], Paul-Sorin Lazăr [2] , Iustin-Atanasiu Pop [2] and Traian-Ionuț Luca [2]**

[1] Management and Economic Engineering Department, Technical University of Cluj-Napoca, 103-105 Muncii Blvd, 400114 Cluj-Napoca, Romania; nastaiancuta@yahoo.ro (A.-N.R.); alinacorda@yahoo.com (A.-M.C.); ainastasoiu@yahoo.ro (I.-A.N.)

[2] Faculty of Business, Babeș-Bolyai University, Strada Horea 7, 400038 Cluj-Napoca, Romania; paul.lazar@ubbcluj.ro (P.-S.L.); iustin.pop@ubbcluj.ro (I.-A.P.); traian.luca@ubbcluj.ro (T.-I.L.)

\* Correspondence: maierdoru@yahoo.com; Tel.: +40-741075576

**Abstract:** This study uses cross-section regressions and spatial econometrics techniques to identify determinants of rural development project implementation based on the Common Agriculture Policy (CAP) of the European Union. For this, we use 40 Romanian counties. Results show that agricultural land abundancy and land concentration degree are significant positive factors. On the contrary, the local human development level is a negative determinant, low values for this factor being an incentive to compensate the lack of own resources through European funding. No significant effects of the average salary or population density were depicted. Spatial analysis indicates contagion and diffusion processes for fund accession through projects. This behavior is like that in other financial sectors, in which human behavior is a decisive factor, such as the insurance one. A West–East clusterization process is identified for the total project value, conditioned by the identified factors.

**Keywords:** EU funds; rural development; Romania; cross-section regressions; spatial econometrics

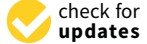



## 1. Introduction and Literature Review

Contemporary societies are increasingly confronted with natural tendencies of high heterogeneities in respect to individual productivity and, implicitly, of population incomes. This situation is due to the transition trends from predominantly physical work to the intellectual one. However, this transition is not made without certain negative consequences. One of these is the danger of poverty of people who have lower income levels. There is, therefore, an obligation for national governments to make social corrections. One of these corrective measures is to ensure a decent standard of living for low-income citizens, to allow them access to goods and services of strict necessity. A main goal in this regard is to finance agriculture by increasing the incomes of agricultural producers, which should influence prices of agricultural products. Consequently, many national governments are acting through economic measures to increasingly support agriculture and rural development [1]. These measures do not only have a social role but also the maintenance of economic balances between the productive economic branches. Agriculture is a sector with naturally limited economic growth due to both extensive factors (limited land areas) and intensive factors, such as productivity, which cannot increase indefinitely [2]. Vulnerabilities due to meteorological whims are also added [3]. We can also mention the rigid supply of food and agricultural products, while the demand is much more volatile, but with a deep upward trend [4]. All the problems already mentioned have the consequence of a much lower natural development of the agricultural sector than other productive activities [5]. Consistent economic theories argue that agriculture has a naturally uncompetitive productivity compared to other economic sectors and consequently needs to be supported by appropriate policies to have optimal functionality [6,7].

All these considerations have already become classics in the economic literature. More recently, arguments in the field of sustainable development and ecology have been added. There are, therefore, financing and subsidy needs to encourage the protection of the soil and other natural resources, to benefit in the long run from agricultural products at a quality that does not decrease over time [8]. It is also worth mentioning the need to support agriculture and rural development, because it is not only desired to obtain goods for direct use. We can also talk about the role of providing externalities and public goods. Among them, we can refer to the extensive supply of jobs that reduce the pressures in the urban environment, to the offered landscapes, healthy environment, etc. [9]. All these concepts regarding the theoretical framework of the externalities provided by the rural environment and agriculture are described in detail in the economic literature [10–12]. Economists argue that the agricultural sector is not just a supplier of specific goods. These have, unlike many other goods and services, a social character, with a social demand that must be covered [10,11,13]. Satisfying social demand is more stringent as interventions in the agricultural sector through project funding. Subsidies are made to ensure food security, whether it refers to a national or international framework [14].

### 1.1. Economic Studies on Financing for Cohesion and Convergence

Policies that have incentives as a theoretical basis and mode of action are based on economic interventions. These have the role of creating comparative advantages or disadvantages for certain behaviors through prices or costs. In policies that use incentives, the causal relationships between certain economic and social phenomena, processes or indicators are exploited. Although causal mechanisms are universally valid due to the non-coercive nature of funding policies, their consequences can be very different from one geographical area to another [5]. There are certain conditions related to administrative specificities, the legal economic framework, the already existing human and material potential, the learning and adaptation processes, etc. As a result of these differences in the values of the mentioned factors, the benefits resulting from the rural development funds could be very different from one administrative unit to another. The economic literature does not have unanimous opinions on the effects of European funding on the processes of economic convergence. Some studies consider that inadequately accessed funding can increase the heterogeneity of economic development [15]. It was also highlighted that exact geographical areas with a more accentuated rurality have accessed less funds that allow productivity increases by incorporating innovative technologies. Instead, other research shows convergence effects because of funding through European Union cooperation mechanisms [16,17].

For a long time now, there have been hopes within the European Union that funds from all regional development programs will bring about a uniformity in the level of development and living standards of the population in different countries and regions. To a large extent, however, hopes were dashed. There are concerns in the literature to determine the major factors of this partial failure. Some authors [18] argue that the lack of productivity of subsidies caused this phenomenon. Instead, there are other opinions that claim [19] that there are phenomena of increasing heterogeneity within countries that cause these disparities at the European level. Another explanation considered refers to the ability to use these funds. Through this perspective, it is considered that, in regions with high human and material development, European financing is absorbed and capitalized in a superior way, accentuating the disparities [5]. To highlight all these mechanisms, it is necessary first to emphasize through sets of very exact indicators the existing situations in territorial profile. To have a clearer picture, the territorial units should be mapped from this perspective at the lowest possible territorial level. However, such evaluations are quite rare in empirical studies. Rather, sectoral situations are assessed, for example, regarding the labor market [20] or the distribution of population income [21]. The lack of exhaustive studies on all economic components is explainable by the complexity of such an approach.

The main goal of the structural funds is to sustain cohesion and economic convergence of the Member States. The value of the allocated funds increased significantly in the late 1980s and then after 2004, as the number of EU countries increased. The issue of optimal allocation and impact assessment of funding has been raised. From the point of view of economic mechanisms, these funds aimed to promote convergence through increasing productivity. The first empirical studies on factors and processes that promote or inhibit convergence included countries that joined the EU in the 1970s and 1980s: Ireland, Greece, Spain, and Portugal. For the 1970s, for example, for Ireland, labor market imbalances were identified as a negative factor [22]. For the 1975–1985 period, in the case of Spain, too relaxed monetary and fiscal policies were identified as inhibitors, and for Greece, the inefficiency of the administrative system. In the 2000s, literature primarily addressed the case of Eastern European countries. For example, the regional institutional impact of preparing for the implementation of the Structural Funds in Poland and the Czech Republic was examined [23]. It was found that, during the pre-accession funding period, there were no major improvements in administrative relations at regional level. Instead, with the accession to the EU, the participation of regional authorities in the processes of absorption, distribution and monitoring of funds has changed decisively. Topics related to the effects of the European Structural Funds are very diverse. Despite the various historical, political, socio-economic, and cultural contexts, convergences have been highlighted on the basic concepts of spatial planning between the cases in North-West Europe and those in Lithuania, Latvia, and Estonia [24]. The content and principles of spatial planning strategies have a degree of similarity that is constantly growing, but implementation, monitoring, and review have a much lower degree of convergence. After 2007, the interest in studying the effects of European funds has shifted primarily to the countries that have most recently joined the European Union: Croatia, Bulgaria, and Romania. For Romania and Bulgaria, several problems have been identified [25] that slow down the convergence process. Both countries face major implementation problems, primarily due to administrative inefficiencies. These often led to delays in absorption of funds and financial irregularities. The pressure to use the allocated funds has led to changes in priorities towards projects that are easier to implement, rather than projects that increase convergence. Due to major deficiencies in infrastructure development, a large part of the funds was directed to transport and construction projects and a relative neglect of innovation and employment. The economic literature is therefore concerned with a wide variety of issues regarding the absorption of European funds and their effects, as well as with national and regional specificities that influence the mechanisms for allocating funding. We can, therefore, speak of specific determinants, depending on the type and destination of a certain financing. Particular aspects can also emerge from the study of the distributions and effects of funds for agriculture and rural development.

Instead, there are recent economic concerns that want to clarify the effects of structural funding, not only on the rural environment but also on local and regional economic development [26]. Rural development is focused on several very precise aspects: agricultural production, the social and economic level of the inhabitants, the sustainable protection of the environment against the effects of agricultural activities, etc. Instead, local and regional development has a broader vision, given the externalities that agricultural and agro-industrial activities manifest on the production of goods and services, the construction sector, the development of innovative technologies [27].

The two concepts, rural and local development, are interconnected when there are more important local and regional connections between agriculture and other economic branches. Moreover, an important bridge is represented by new technologies [5]. Very often, these technologies targeted by rural development funds aim to improve the quality and safety of agricultural products, increase productivity in technological processes, or improve the working conditions of employees. The development of technologies takes place, in most cases, within zonal groups of highly specialized productive activities. These are usually found where there are regional and local organizational developments specific to the developed contemporary economy [28]. If there are, therefore, combined strategies of

rural and regional development, from their pooling special comparative advantages result. These are manifested by higher levels of employment in the labor market, especially for highly qualified trades, increasing the complexity of the production of goods and services. There is also an increase in the level of socio-economic development of the population and the maintenance of citizens in rural areas.

### 1.2. Common European Framework for Financing Rural Development

In a historical context, with the changes in optics regarding the support of agriculture as an economic sector, the ways in which agriculture was supported have also evolved. The first organized form of support at an international level was the Common Agricultural Policy (CAP) of the European Union [5]. In the 21st century, support interventions have taken better and better organized forms and focused on two axes. The first axis refers to actions on the markets and prices of agricultural products. The second axis finances structural projects whose main objective is the economic and social development of rural areas. The specific institutions of the European Union have developed concrete strategies, which aim at several distinct objectives, directly related to rural development. Among them, we mention the increase in productive capacities and the quality of farm products, protection of the environment as a result of agricultural activities, protection and improvement of living standards of villagers, including by creating jobs, etc. [29]. The EU's Common Agricultural Policy has undergone a major transformation since 2013. The second axis of intervention has been declared a priority. As a result, policies targeting the rural area have focused primarily on complex, multifunctional development. Emphasis was also placed on the capacity of agriculture and related industrial activities to provide public goods [5]. Time intervals have been defined in which each Member State can propose its own policies and programs for rural development (RDP). They are submitted for approval and funding to the European Agricultural Fund for Rural Development (EAFRD). There is a great deal of freedom of the national administrative authorities in establishing their own intervention mechanisms, but under the restrictions of being included in the CAP objectives. Practically, each government becomes directly responsible for the efficiency of the measures taken and the correct mix between them. The effectiveness of the measures of the second axis is therefore conditioned by the national governmental capacities to define coherent policies and measures and of their correct implementation [30]. Accessing funds through the second axis is not mandatory but optional. Funding is granted based on requests from potential beneficiaries. The applications materialize in the form of fundable projects. There are also incentive measures in this mechanism. Tools for action on the prices of agricultural products and their derivatives are used to correctly direct the actors involved in these activities [31].

The budget allocated by the European Union through EAFRD for the period 2014–2020 is around EUR 95 billion [European parliament]. Following the regulations on the implementation of the CAP of 23 December 2020, the RDPs have been extended under certain clearly specified conditions for 2021 and 2022. In this additional interval, the RDPs will receive EUR 26.9 billion from the EAFRD budget for 2021–2027. In addition, EUR 8.1 billion will be allocated, money from the European Union's recovery instrument for the next development period. Due to this expansion, many of the projects and schemes included in the RDP will continue to run until the end of 2025 [32].

### 1.3. Determinants of the Regional Distribution of European Funds for Rural Development

Although not completely absent from the literature, studies on the factors that determine the spatial distribution of development funds are quite few, especially in the case of those for rural development. Probably the phenomenon is due to the fact that, at the member state level, the distribution is made administratively, by negotiating with the member states, and not by economic mechanisms to attract funding. Instead, at the regional and local level, the distribution of funds is much more likely to be influenced by certain local economic, social, political, or administrative determinants. There are academic

concerns [33] for defining typical directions for the development of local communities. The research highlights several social, economic, and political indicators that can be used as explanatory variables of the spatial distribution of development funds. Four basic profiles of local development dynamics are also identified. The territorial distribution of European funds for the development of countries and regions was analyzed globally for several types of financing [34]. At the level of the EU component states, an important determinant is the negotiation process. Results show that this factor is far from being able to explain alone the distribution at country level, and at regional and local level it is insignificant. Instead, several economic and institutional indicators are highlighted, along with unique characteristics of the electoral competition. For the particular case of rural development, the value of funds attracted at a regional level was estimated in order to develop agriculture and rural areas [35]. The relationship between the level of absorption of European funding and several socio-economic indicators calculated at local and regional level was examined: the migration balance, the percentage of localities within a commune with access to public transportation, the proportion of non-agricultural business entities in the total number of businesses in the commune, the number of pensioners receiving Agricultural Social Insurance Fund, the share of taxpayers, and the proportion of the commune's own income in overall revenue. The field's literature [5] also highlighted some correlations between the frequency of applications for rural development measures of the Common Agricultural Policy and some explanatory variables that measure the development of agriculture and the rural area in a given region. The empirical study examines Poland's example and shows that government policies prioritize funds that primarily focus on developing farms and areas with higher natural agricultural potential, with high risks of increasing regional development disparities. Another study of regional distributions [36] shows, in the case of Slovakia, that the allocation of European agricultural policy funds varies dramatically from one region to another. Variability cannot be explained only by local development indicators. The role of local public administration performance is also highlighted, as well as the indirect effects in knowledge, financial means, and social relations. Econometric models that have highlighted these causalities use panel data and a Durbin spatial model.

*1.4. Rural Development in Romania through European Union Funding*

Romania became a member of the European Union in 2007. Like other countries, especially in Central and Eastern Europe, it has benefited from pre-accession and post-accession funds for economic cohesion and convergence. During the pre-accession period, Romania received assistance through three financial instruments, PHARE, ISPA, and SAPARD, which were joined by community programs. Regional, rural, agricultural, and environmental development programs based on national regional development policies were funded, thus ensuring Romania's transition to the structural funds system. PHARE has supported Central and Eastern Europe in its evolution toward a democratic society and a market economy, focusing on two aspects, namely institutional development and investment. ISPA has provided financial support for investments in environmental protection, transportation, and legislative harmonization. SAPARD has supported candidate countries in addressing structural reform in the agricultural sector and other areas related to rural development. After joining the EU, Romania benefited from European structural funding in two programming periods, respectively, 2007–2013 and 2014–2020. In the first period, through the operational programs, the first three economic components financed were transport, environment, and human development. In the second period, agriculture and rural development attracted more attention, the financing of these components being in second place, immediately after transport. Through PNDR 2014–2020 (National Rural Development Program), EUR 8.1 billion was granted for the economic and social development of the rural area in Romania, whereas the total amount on rural development was EUR 95.3 billion [37]. Per capita, the financing for Romania was EUR 413.5, almost double the EU average of EUR 213.7. The result seems to be in line with the EU's policy of giving priority support to poorer countries and regions. The positive financing discrimination is further blurred if we

report the total funds allocated to the rural population, Romania having a share of 46% of citizens living in villages, compared to the EU average of 25.7%. The situation is similar in terms of land distribution, with a percentage of 95.2% of rural land area compared to 81.3% in the EU. Despite the higher urbanization, Romania benefitted from a financing of EUR 898.9 per rural inhabitant in comparison with an EU average of EUR 831.5. It is interesting to follow the financing related to the available agricultural and arable land, which is exactly opposite. For Romania, a financing from European funds of EUR 605.7 per hectare of agricultural land was achieved, below the EU average of EUR 928.3. If we report the financing of rural development to the available arable land, we obtain almost similar results of EUR 948.5 per hectare for Romania and EUR 958.2 for the EU average. These statistics reveal at least one interesting aspect. Funding relative to the rural population of poorer countries can help them compete with similar people in more developed countries. On the other hand, the lower financing per hectare of agricultural land in Romania compared to the EU is likely to fail to lead to a competitive price of agricultural products, for example. Of course, all these considerations refer to national values and averages. Within each country, there are financing disparities to the advantage of economic convergence, depending on administrative and socio-economic mechanisms.

As already mentioned, each member country of the European Union establishes its own intervention policies for rural development. In Romania, the most important attributions in defining the objectives and managing the financing processes belong to the Ministry of Agriculture and Rural Development (MADR) and to some administrative structures subordinated to it [38]. The most important institutions subordinated to this ministry that have attributions regarding the financing of agricultural activities and for rural development are Agricultural Payments and Intervention Agency (APIA) and Agency for Financing Rural Investments (AFIR). The Ministry has as an attribution the realization of the National Rural Development Program (PNDR) in accordance with the objectives established by the Common Agricultural Policy [39]. Like the policy of the European Union, the financing of agriculture and rural development in Romania is carried out on two axes.

APIA manages the activities related to the first axis. As a structure, the agency has a headquarter, county territorial headquarters (42), and locations of local importance (266). EU funding is being carried out through APIA for the materialization of some support measures that come as funds from the European Agricultural Guarantee Fund (EAGF). Among the attributions of these agencies are the management of financing from national and European state funds, the management of payment requests from possible beneficiaries of payments, and the approval of the quality and conformity of agricultural and agri-food products within import and export activities, informing producers and traders about the activities and programs carried out.

AFIR manages the activities corresponding to the second axis. The existence of this agency is justified by the obligation of each EU country to nominate a specialized institution for these types of rural development programs, authorized from a legislative point of view to manage payments and absorb European funds. The major obligation of AFIR is the administrative and financial implementation of the European Fund for Agriculture and Rural Development (EAFRD).

The Rural Development Program (RDP) for Romania was accepted and scheduled for funding by the European Commission on 26 May 2015. The last changes occurred on 26 January 2021. The national program defined the specific priorities to use the amount as appropriate as possible of EUR 9.5 billion of European funding available for a period of 7 years, i.e., between 2014 and 2020 (EUR 8.1 billion from the EU budget, including EUR 112.3 million transferred from CAP direct payments and EUR 1.34 billion from national co-financing) [40].

*1.5. Research Objective and Motivation*

Considering the cited literature, it is important to identify the extent to which rural areas in certain regions or counties are involved in absorbing the funding allocated to

them based on the convergence and cohesion policies of the European Union. There are few studies of this type conducted, for example, for Poland [33,35] or Slovakia [36], but for Romania they are missing. However, studies that focus on distribution within the regions of a country are quite rare, preferring those that analyze the distribution and effects between countries. However, the distribution mechanisms are different. For example, the allocation of funding for each country is achieved through negotiations between the EU and member countries. Instead, within countries, there is often free competition through applications for funding. Even studies on funding flows of a certain type, such as those for rural development, are less studied, with empirical research preferring, in most cases, to assess issues related to total funding. The main purpose of our study is to determine if there are socio-economic mechanisms and effects of diffusion or spatial contagion that intervene in the geographical distribution of funds coming from the second axis, from EAFRD, on the Romanian territory. We investigate the influence of several variables at county level that can interfere in the spatial allocation model.

The allocation of funds between EU countries is at least partially technically and economically unjustified if negotiation processes take place. Spatial grouping of structural funds is often observed in member countries, and clusters do not necessarily reflect the spatial dependence of criteria officially defined by cohesion policies [41]. For example, regional and local governments may influence the distribution of funds between regions and localities in a country through electoral or political pressures or through different fund management capabilities and different co-financing capabilities. Beyond the administrative aspects, there are social and economic factors that may or may not favor absorption. Among them, some determinants of agricultural nature can be identified. In regional profile, the share of agricultural and arable land, the number of farms and their average size, and the degree of land concentration may differ substantially. Interestingly, this class of factors has been neglected in the literature, even in the few studies on the distribution of European funds for rural development. Rather, some socio-economic determinants identifiable at the level of regions or localities were researched, such as the migration balance, the percentage of localities within a commune with access to public transportation, the proportion of non-agricultural business entities in the total number of businesses, the number of pensioners receiving Agricultural Social Insurance Fund, the share of taxpayers, and the proportion of the commune's own income in overall revenue [35]. Due to the high collinearity of these variables in these studies, they opted for descriptive statistics or for the separate introduction of socio-economic factors into regressions, but these technical options may introduce biases in the results. In our study, we prefer to evaluate local human development in a region or county using a composite index.

As noted in empirical research on the role of administrative factors [36], spatial spillovers can also manifest for agricultural or socio-economic factors, for which appropriate econometric methods are spatial regression models. The spatial distributions of social and economic variables reflect their spatial dependencies. In addition, as in the case of mimetic learning processes, the theoretical knowledge and skills necessary for efficient application for accessing funds are more easily transmitted in neighboring areas. In the case of our research, these spreads can be manifested both for local governments and for rural communities and farmers. An additional argument is given by the possibility of collaboration of neighboring communities or companies to carry out larger joint financing projects. This behavior is encouraged by the legal criteria of rural development funds. Although these mechanisms of diffusion are generally known, the literature is very poor in evaluating them empirically, especially through spatial econometrics [36].

In our research, we perform cross-section and spatial econometric analyses of the determinants of accessing funds for rural development within the Romanian counties. We use data from 2014–2020, the second allocation period after joining the European Union. In summary, we can say that we explain the role of determinants in the structure of agricultural land, local human development, and the effects of spatial diffusion on the number and value of applications for accessing European funds for rural development. Conclusions

can thus be drawn on the efficient allocation of funds, i.e., whether they finance the less developed regions as a matter of priority, to meet the needs of economic convergence.

Several indicators used as a proxy for the granted projects are considered: the number of projects financed from a certain territorial unit, the total value financed from a county, and the average value of the projects that have been approved for financing. The analysis is performed both by classical regression models and by spatial econometrics. As a result of the study, regional policy findings and recommendations are made.

## 2. Materials and Methods

### 2.1. Working Hypotheses

Based on the study of the cited literature and on our own reasonings regarding the possible logical causal relations regarding the geographical distribution of the number and value of the European funded projects, we formulate the following working hypotheses. To avoid possible misunderstandings of the terms used in this study, an application for funding from a farmer or an economic entity is referred to in the article as a project.

**Hypothesis 1 (H1).** *There is a county level variation in the number and value of projects depending on the number of farms and available agricultural areas.*

Although there is no consolidated economic theory in this respect, we can expect a larger number of farms in fact correlated with the number of landowners to generate more funding requests and, consequently, a larger number of projects. In addition to the fact that this is a significant effect, a higher number of farms and farmers arelikely to generate a higher level of competition.

**Hypothesis 2 (H2).** *The number and value of projects depend on the degree of concentration of agricultural land.*

The mechanism considered by this hypothesis concerns some economic aspects. Larger farm areas encourage access to development funds. First, in the access process there are some fixed costs, which are easier to cover for large farms. Second, a larger community is more likely to have more competent employees with better access to funding projects.

**Hypothesis 3 (H3).** *Human development is a negative determinant of accessing funds for rural development.*

Lower human development can be a serious incentive to access development funds, due to the poorer quality of life in that region. We can accept that human development is a very general concept, difficult to capture by a single variable. In our study, we will use as a proxy a composite index of local human development, inspired by the HDI (Human Development Index), namely the Local Human Development Index (LHDI), developed for Romania, for the World Bank [42].

**Hypothesis 4 (H4).** *There are contagion and diffusion phenomena in the processes of accessing funds.*

These phenomena are often encountered in economic processes involving human financial behavior, for example, in insurance. The mimetic effects, in which you act in a certain way because you see that this is what the neighbors do, are often signaled in economic and administrative activities. In addition, there may be learning processes, the way of managing the submission processes and project management can be learned from people with whom you come into contact more often.

### 2.2. Data and Methodology

The data used in this study were collected from official public sources: INSEE (Romanian National Institute of Statistics) [43], AFIR (Agency for Financing Rural Invest-

ments) [39], and MADR (Ministry of Agriculture and Rural Development) [38]. The data on the number and value of projects cover the time interval 2014–2020. The other economic explanatory variables refer to the middle of this interval, i.e., 2017. A summary of the variables used, with the abbreviated name used in regressions, explanations, and some descriptive statistics can be found in Table 1. Of the 42 counties of Romania, only 40 were used in the empirical study. Two counties, respectively, the capital Bucharest and its adjacent area, are not considered because they mainly represent urban areas.

**Table 1.** The variables included in the analysis.

| Variable Name | Explanations | Median | Mean | St. Dev. | Coef. Var. (%) |
|---|---|---|---|---|---|
| NB_PROJECTS | Number of active contracted projects [1] (between 2014 and 2020) at the level of a county. | 513 | 572 | 420 | 73.4 |
| VAL_PROJECTS | Value of active contracted projects (between 2014 and 2020) in millions RON at the level of a county. | 14.18 | 17.22 | 14.05 | 81.6 |
| PROJECT_VALUE | The average value of a project in a county. It is calculated as the ratio between the variables VAL_PROJECTS and NB_PROJECTS (in thousand RON). | 29.1 | 30.3 | 8.06 | 26.6 |
| NB_ARABLE_HOLD | Number of arable land holdings in the county. | 64,179 | 68,471 | 29,282 | 42.8 |
| NB_AGRIC_HOLD | Number of agricultural land holdings in the county. | 89,326 | 92,373 | 34,800 | 37.7 |
| AREA_ARABLE | The area of agricultural land in the county (in hectares). | 166,905 | 206,224 | 128,078 | 62.1 |
| AREA_AGRIC | The area of agricultural land in the county (in hectares). | 331,121 | 331,091 | 104,165 | 31.5 |
| RATIO_ARABLE | The ratio between the total arable area in a county and the number of holdings with arable land in the same county. | 2.506 | 3.828 | 3.69 | 96.4 |
| RATIO_AGRIC | The ratio between the total arable area in a county and the number of holdings with arable land in the same county. | 3.567 | 4.144 | 2.11 | 50.9 |
| DENSITY | Population density in the county (inhabitants per square kilometer), in 2017. | 74.0 | 77.1 | 26.4 | 34.3 |
| WAGE | Medium salary in the county (calculated in the middle of the interval 2014–2020, i.e., June 2017). | 1950 | 2028 | 219 | 10.8 |
| LHDI | The index of local human development measures, the total capital of localities, looking at four dimensions: human capital, health capital, vital capital, and material capital. | 0.621 | 0.6291 | 0.08 | 12.7 |
| Dummy variables | The region (NUTS 2) in which the county is located is indicated. The variable has the value 1 if the county is in that region and the value 0 otherwise. | | | | |
| SOUTH_WEST | South-West region. | | | | |
| WEST | West region. | | | | |
| NORTH_WEST | North-West region. | | | | |

**Table 1.** *Cont.*

| Variable Name | Explanations | Median | Mean | St. Dev. | Coef. Var. (%) |
|---|---|---|---|---|---|
| CENTER | Center region. | | | | |
| SOUTH | South region. | | | | |
| SOUTH_EAST | South-East region. | | | | |
| NORTH_EAST | North-East region. | | | | |
| W_VAL_PROJECTS | Spatial lag of VAL_PROJECTS, which is the weighted average of VAL_PROJECT for the neighbors of each spatial unit, as defined by the spatial weights matrix. | | | | |

Source: Authors' construction using data provided by INSEE [43], AFIR [39], and MADR [38]. [1] To avoid possible misunderstandings of the terms, an application for funding from a farmer or an economic entity is referred to in the article as a project.For the variables NB_PROJECTS, VAL_PROJECTS, and PROJECT_VALUE, only those applications for funds that have been accepted for financing and for which payments have been made are taken into account.

The first step of the analysis consists of the descriptive assessment of data. However, since we are dealing with information at the county level for Romania, besides the classical descriptive statistics, we also employed the visual analysis based on maps. More specifically, we constructed the quartile maps for the dependent variables, on one hand, and the differences between the estimated values and the real ones (actually, the errors of the estimation process), along with PROJECT_VALUE, on the other hand. For the actual spatial assessment, we construct and employ the spatial weights matrix (W) in the queen contiguity form, based on actual frontiers. The quartile maps point out spatial clusterization processes and we further test H6 with the help of the spatial autocorrelation analysis and the global spatial autocorrelation coefficient of Moran, Moran's I, just like Mare et al. [44]. This evaluates the relationship between the value of each spatial unit and its neighbors. A positive coefficient confirms the clusterization process and is also indicative of significant contagion and diffusion.

Finally, the most efficient regressions from the classical perspective are respecified in the spatial form. We start with the OLS method to which we add the spatial weights matrix. Post estimation spatial diagnosis tests confirm the need for spatial effects in the form of spatial lag of the dependent (spatial autoregressive model SAR), but only for the respecification of model 6. Consequently, we have models 10 (the OLS with spatial weights matrix) and 10* (SAR). The SAR model includes the spatial lag of the dependent variable. This is the weighted average value of the dependent in the neighbors of each spatial unit (namely county) analyzed. Consequently, it allows for the comparison of the value in each county with the average one in the neighbors, allowing for the emphasis of similar or different behaviors in respect to European funding, in space. *p*-values of the spatial diagnosis tests are provided along with the R2 values of the models. Analyses were conducted in STATA 14 and GeoDa 1.14.

### 3. Results and Discussion

Descriptive statistics show a very high variability (through the Pearson coefficient of variation) of the variables that describe the number and value of projects. The situation is very similar in terms of the variables that describe the agricultural areas and the number of farms. Instead, the variability of economic variables is much smaller and may suggest that they can only explain to a limited extent the number and value of projects.

A reduced variability is also found in the average value of the projects, suggesting a certain uniformity on the national territory. However, descriptive statistics cannot show the territorial distribution of the analyzed phenomenon. For this, we show in Figures 1 and 2 the distribution by counties of the situation of the grants for rural development.

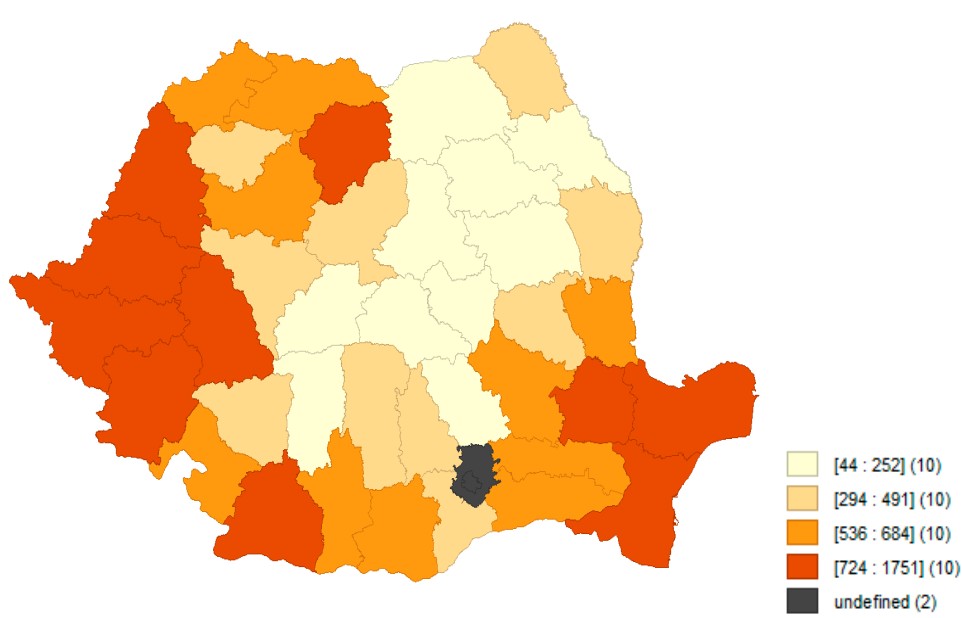

**Figure 1.** Distribution of the number of active contracted projects (NB_PROJECTS) by counties. The legend indicates the limits of the quartile intervals for the variable NB_PROJECTS and the number of counties in each quartile interval. Source: authors' construction in GeoDa 1.14.

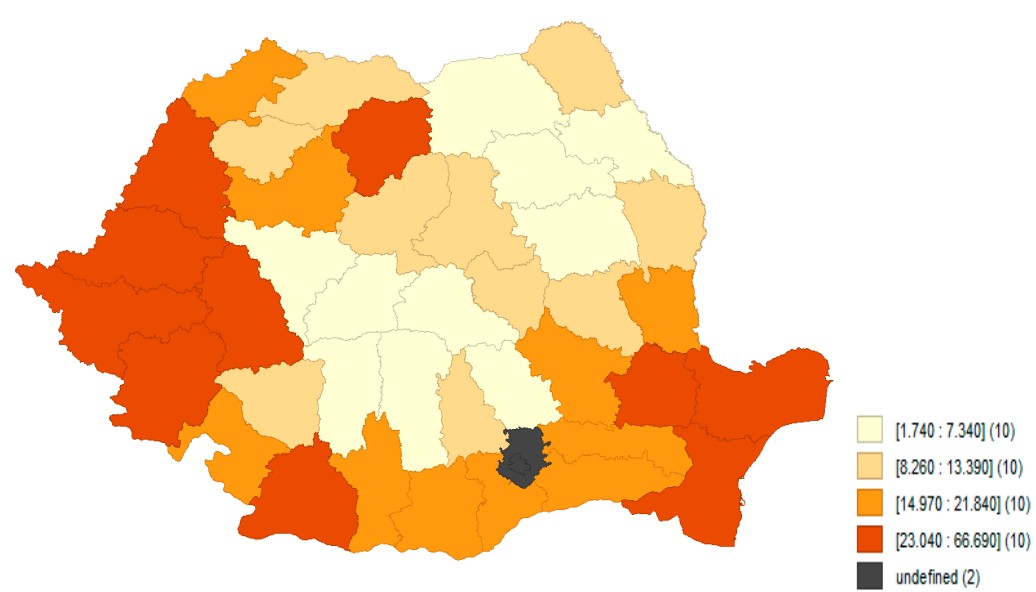

**Figure 2.** Distribution of the number of active contracted projects (VAL_PROJECTS) by counties. The legend indicates the limits of the quartile intervals for the variable VAL_PROJECTS and the number of counties in each quartile interval. Source: authors' construction in GeoDa 1.14.

Maps in Figures 1 and 2 show that both dependent variables have a similar spatial positioning, with the highest values clustered in the West and South-East parts of Romania. They are followed by southern counties, which, by default, are agricultural areas. However, in comparison with the most performant group, these counties are among the least developed in Romania, lacking the resources for writing and applying to rural structural funds. Consequently, we expect spatial factors to be significant in the regressions, along with the development level.

Cross-section regressions model the causal relationships formulated by hypotheses. To evaluate the determinants of the number of projects, we used several alternative models (models 1 to 4, Table 2).

**Table 2.** Results of OLS regressions on the dependent variable NB_PROJECTS (coefficients and *t*-stat).

| | Model 1 | Model 2 | Model 3 | Model 4 |
|---|---|---|---|---|
| NB_ARABLE_HOLD | 0.003 (0.83) | | | |
| NB_AGRIC_HOLD | | 0.003 (1.01) | | |
| RATIO_ARABLE | | | *** 54.76 (3.12) | |
| RATIO_AGRIC | | | | *** 115.7 (3.68) |
| DENSITY | −2.423 (0.83) | −3.074 (−0.93) | 0.965 (0.37) | 1.753 (0.69) |
| WAGE | 0.374 (1.12) | 0.395 (1.18) | −0.029 (−0.10) | −0.071 (−0.25) |
| LHDI | ** −2064 (−2.65) | *** −2176 (−2.74) | * −1261 (−1.79) | * −1323 (−2.00) |
| NORTH_WEST | ** 468.2 (2.14) | ** 482.9 (2.23) | ** 447.5 (2.95) | ** 346.4 (2.11) |
| CENTER | 147.5 (0.53) | 170.9 (0.63) | 48.03 (0.25) | −156.1 (−0.83) |
| WEST | ** 748.9 (2.68) | *** 756.3 (2.79) | *** 609.0 (2.80) | 358.6 (1.62) |
| SOUTH_WEST | * 388.1 (1.81) | * 400.5 (1.87) | ** 416.2 (2.21) | ** 415.8 (2.32) |
| SOUTH | 214.0 (1.03) | 186.9 (1.00) | −42.87 (−0.26) | 6.854 (0.04) |
| SOUTH_EAST | *** 750.1 (2.80) | *** 766.9 (2.98) | 294.0 (1.45) | 183.4 (0.90) |
| NORTH_EAST | reference | reference | reference | reference |
| constant | 768.4 (1.10) | 762.0 (1.12) | 924.4 (1.67) | 793.7 (1.49) |
| $R^2$ | 0.562 | 0.567 | 0.664 | 0.695 |

*, **, *** statistically significant at 10%, 5%, and 1% level. Source: authors' calculation in STATA 14.

Variables that describe the number of agricultural holdings (or with arable land) and the ratios between the agricultural area and the number of holdings are very correlated with each other (Table 3). Consequently, they are alternately introduced into models as explanatory variables.

**Table 3.** Correlation matrix (Pearson) between variables.

| | (1) | (2) | (3) | (4) | (5) | (6) | (7) | (8) | (9) | (10) |
|---|---|---|---|---|---|---|---|---|---|---|
| NB_PROJECTS (1) | 1.00 | | | | | | | | | |
| VAL_PROJECTS (2) | 0.95 | 1.00 | | | | | | | | |
| NB_ARABLE_HOLD (3) | −0.20 | −0.26 | 1.00 | | | | | | | |
| NB_AGRIC_HOLD (4) | −0.24 | −0.31 | 0.94 | 1.00 | | | | | | |
| AREA_ARABLE (5) | 0.66 | 0.62 | 0.02 | 0.03 | 1.00 | | | | | |
| AREA_AGRIC (6) | 0.71 | 0.65 | 0.04 | 0.04 | 0.86 | 1.00 | | | | |
| RATIO_ARABLE (7) | 0.58 | 0.57 | −0.52 | −0.44 | 0.75 | 0.59 | 1.00 | | | |
| RATIO_AGRIC (8) | 0.63 | 0.64 | −0.70 | −0.70 | 0.53 | 0.58 | 0.83 | 1.00 | | |
| DENSITY (9) | −0.25 | −0.33 | 0.40 | 0.54 | 0.07 | −0.01 | −0.13 | −0.32 | 1.00 | |
| WAGE (10) | 0.02 | 0.04 | 0.06 | −0.02 | 0.06 | 0.17 | 0.02 | 0.13 | 0.44 | 1.00 |
| LHDI (11) | −0.42 | −0.42 | 0.11 | 0.15 | −0.42 | −0.21 | −0.37 | −0.20 | 0.23 | 0.38 |

Source: authors' calculation in STATA 14.

First of all, we note that NB_ARABLE_HOLD and NB_AGRIC_HOLD are not statistically significant, although it was logically expected that a larger number of farms in a county would generate more projects. Instead, the variables RATIO_ARABLE and RATIO_AGRIC are very significant and have the expected positive sign. Consequently, a higher concentration of agricultural land, by creating larger farms, generates a larger number of projects. This is explained primarily by economic mechanisms. Submitting a project is a fixed cost. It has a lower share in the total expenses of a farm if it is higher. Secondly, a larger size of the farm implies more staff, favoring teamwork. We then note that the LHDI variable is significantly negative in any specification. This indicates that lower human development implies a greater need for access to funding, other than from own sources, especially if they are non-reimbursable. Of the dummy variables that indicate the county's membership in a particular region (NUTS 2), some of them (NORTH_WEST and SOUTH_WEST) are statistically significant in any econometric specification. This result indicates that, in these regions, the number of submitted projects would be higher at the same values of the influencing factors. This may indicate greater efforts by regional public authorities in the process of encouraging access to European funds for rural development. Instead, other variables (SOUTH_EAST) lose their significance when the variables RATIO_ARABLE and RATIO_AGRIC appear in the regressions, being more correlated with

them. However, the role of geographical positioning is difficult to highlight by cross-section regressions; it will be better identified by spatial analysis.

Cross-sectional regressions on the total value of projects (VAL_PROJECTS) indicate more complex results (models 5 to 8, Table 4). The role of LHDI seems to be the same, with higher human development indicating a prosperity of society that has more abundant own resources, so it does not strongly encourage access to European funding. Instead, both the more generous agricultural (or arable) area and the concentration of land in larger farms (RATIO_ARABLE and RATIO_AGRIC) are positive determinants of the total value of the financed projects. Population density (DENSITY) appears to be negatively significant. However, the result is not very conclusive, the variable being calculated for the whole county, not exclusively in rural areas. The role of the variables that indicate the region is more difficult to elucidate, the significance varying from one regression to another.

**Table 4.** Results of OLS regressions on the dependent variable VAL_PROJECTS (coefficients and *t*-stat).

|  | Model 5 | Model 6 | Model 7 | Model 8 |
|---|---|---|---|---|
| AREA_ARABLE | *** $5.64 \times 10^{-5}$ (4.18) | | | |
| AREA_AGRIC | | *** $6.4 \times 10^{-5}$ (4.66) | | |
| RATIO_ARABLE | | | ** 1.340 (2.27) | |
| RATIO_AGRIC | | | | *** 2.982 (2.80) |
| DENSITY | * −0.139 (−1.92) | * −0.126 (−1.81) | −0.088 (−1.00) | −0.065 (−0.75) |
| WAGE | 0.009 (−1.05) | 0.010 (1.19) | 0.011 (1.03) | 0.009 (0.92) |
| LHDI | −28.97 (−1.38) | ** −46.09 (−2.45) | ** −46.88 (−1.98) | ** −47.61 (−2.12) |
| NORTH_WEST | ** 11.86 (2.34) | ** 9.870 (2.07) | 8.912 (1.53) | 6.385 (1.15) |
| CENTER | 2.726 (0.48) | 1.496 (0.28) | −1.513 (−0.23) | −6.721 (−1.05) |
| WEST | ** 16.18 (2.58) | ** 12.68 (2.09) | ** 16.53 (2.26) | 10.02 (1.33) |
| SOUTH_WEST | 7.998 (1.48) | ** 10.83 (2.09) | * 10.73 (1.70) | * 10.77 (1.77) |
| SOUTH | 0.536 (0.12) | 3.093 (0.72) | −0.520 (−0.09) | 0.544 (0.11) |
| SOUTH_EAST | *** 15.20 (2.91) | *** 18.19 (3.71) | * 12.01 (1.76) | 8.764 (1.27) |
| NORTH_EAST | reference | reference | reference | reference |
| constant | 9.712 (0.60) | 8.019 (0.51) | 21.53 (1.16) | 18.04 (1.00) |
| $R^2$ | 0.752 | 0.772 | 0.662 | 0.686 |

*, **, *** statistically significant at 10%, 5%, and 1% level. Source: authors' calculation in STATA 14.

A partial conclusion of the study can be deduced so far. Two factors are constantly significant and in line with the expectations. First, it is about the degree of concentration of farms, which positively influences both the number and the value of projects. It is rather a factor related to the ability to submit projects according to the requirements of the authorities managing the funding process. Second, lower human development is a factor related to the need to access funding, in the absence of own resources for development. In short, the main determinants are the ability and need to access funds.

In addition to highlighting some determinants of the number and value of projects, we test the possible existence of phenomena of contagion and spatial diffusion. For this, we compute the spatial autocorrelation coefficient Moran's I. In both cases, the positive and highly significant values for I suggest the existence of contagion and diffusion (Moran's I = 0.296, *p*-value = 0.002 for the number of projects; Moran's I = 0.315, *p*-value = 0.001 for the value of the projects). However, for this, we must go further on and resume the most significant regressions (models 4, 6, and 8, with the inclusion of spatial elements) as specifications in the spatial form. As space is independently included in the analysis, we did not include the region dummies any longer for multicollinearity violation purposes. Results are included in Table 5 (models 9, 10, and 11).

**Table 5.** Results of the spatial model estimations (coefficients and *t*-stat/*z*-stat).

| | Model 9 Dependent Variable: NB_PROJECTS | Model 10 Dependent Variable: VAL_PROJECTS (OLS) | Model 10* Dependent Variable: VAL_PROJECTS (SAR) | Model 11 Dependent Variable: VAL_PROJECTS |
|---|---|---|---|---|
| AREA_AGRIC | | *** $7.31 \times 10^{-5}$ (4.79) | *** $6.79 \times 10^{-5}$ (4.99) | |
| RATIO_AGRIC | *** 106.23 (3.82) | | | *** 3.196 (3.57) |
| DENSITY | −0.642 (−0.27) | *** −0.191 (−3.02) | *** −0.154 (−2.72) | −0.099 (−1.29) |
| WAGE | 0.171 (0.58) | * 0.015 (1.76) | 0.011 (1.50) | 0.13 (1.31) |
| LHDI | ** −1731 (−2.43) | ** −52.92 (−2.52) | ** −44.22 (−2.37) | ** −61.11 (−2.67) |
| W_VAL_PROJECTS | | | ** 0.351 (2.17) | |
| constant | * 924 (1.70) | 11.377 (0.69) | 6.51 (0.45) | 24.57 (1.4) |
| | | Spatial diagnosis tests (*p*-value) | | |
| Moran's I error | 0.140 | 0.100 | - | 0.254 |
| LM lag | 0.292 | 0.033 | - | 0.202 |
| Robust LM lag | 0.593 | 0.048 | - | 0.161 |
| LM error | 0.363 | 0.234 | - | 0.545 |
| Robus LM error | 0.975 | 0.381 | - | 0.401 |
| LM SARMA | 0.573 | 0.070 | - | 0.312 |
| $R^2$ | 0.485 | 0.607 | 0.656 | 0.523 |

*, **, *** statistically significant at 10%, 5%, and 1% level. Source: authors' calculation in STATA 14 and GeoDa 1.14.

When spatial influences are included in the analysis, results remain like the classical ones for the number of projects, with both RATIO_AGRIC and LHDI highly significant. However, while the former's coefficient is lower in the spatial specification, the impact of the LHDI is much higher. It emphasizes the important role played by the development level, just as we hypothesized in H5. In addition, this influence is true not only for the number of projects but also for their value, as shown by models 10 and 11. In respect to RATIO_AGRIC, the impact upon VAL_PROJECTS is higher in the spatial regression from model 11. However, spatial components are well treated by the OLS model with the spatial matrix attached for both models 9 and 11. The spatial diagnosis tests have probabilities higher than the 5% critical level, confirming that there is no need to add additional spatial components.

On the contrary, when RATIO_AGRIC is replaced by AREA_AGRIC, significant spatial effects are emphasized by the LM lag test, pointing out the need to introduce the spatial lag of VAL_PROJECTS in the regression. The latter is significant and with a positive value, confirming the descriptive results visualized on the quartile maps in Figure 2—counties with higher project values are neighboring counties with similar values in respect to the rural development financing lines. In conclusion, for VAL_PROJECTS, the spatial regression confirms both a clusterization of the Romanian counties based on this variable, with similar values neighboring, and significant contagion and diffusion processes taking place. In respect to the factors considered, the impact of the agricultural variable, AREA_AGRIC increases in comparison with the classical regression, just as the one of the population densities. However, while higher agricultural area is leading to more funds used for rural development, the population density restricts it. The negative coefficient is highly significant, rejecting H3.

Obviously, we have not highlighted all the factors that can contribute to accessing European funding. There are some administrative factors that are difficult to quantify. However, they are likely to encourage access to projects, depending on the competencies and motivation of the regional regulatory authorities. To highlight these aspects, we again resort to spatial distributions with the help of maps. We consider that each county has a certain potential for accessing funds, conditioned by the main influencing factors identified: RATIO_AGRIC, AREA_AGRIC, and LHDI. We thus practically consider the motivational factor and the ability factor. Ignoring the other factors, insignificant or with ambiguous influence, we estimate for each county the normal value (estimated from regression functions, Table 6) of the number and value of projects. The coefficients are

taken from Model 4* and Model 8* (Table 6), which are simplified versions of Model 4 and Model 8 (by including only statistically significant factors in the regressions).

$$NB\_PROJECTS\_ES = 112.3 \cdot RATIO\_AGRIC - 1572.2 \cdot LHDI + 1095.4$$

$$VAL\_PROJECTS\_ES = 0.0000513 \cdot AREA\_AGRIC + 2.394 \cdot RATIO\_AGRIC - 45.82 \cdot LHDI + 19.15$$

**Table 6.** Results of simplified OLS regressions on the dependent variables NB_PROJECTS and VAL_PROJECTS (coefficients and *t*-stat).

|  | Model 4*<br>Dependent Variable:<br>NB_PROJECTS | Model 8*<br>Dependent Variable:<br>VAL_PROJECTS |
|---|---|---|
| AREA_AGRIC |  | *** $5.13 \times 10^{-5}$ (2.85) |
| RATIO_AGRIC | *** 112.3 (4.67) | *** 2.394 (2.70) |
| LHDI | ** −1572.2 (−2.49) | ** −45.82 (−2.38) |
| constant | ** 1095.4 (2.54) | 19.15 (1.36) |
| $R^2$ | 0.480 | 0.550 |

**, *** statistically significant at 5%, and 1% level. Source: authors' calculation in STATA 14.

In relation to these estimated values, it can be assessed whether a county is underperforming or overperforming, conditioned by the influencing factors. For this purpose, we calculate the differences between the real values of the number and value of the projects and those estimated by regressions.

$$DIFF\_NB\_PROJECTS = NB\_PROJECTS - NB\_PROJECTS\_ES$$

$$DIFF\_VAL\_PROJECTS = VAL\_PROJECTS - VAL\_PROJECTS\_ES$$

To visualize territorially the geographical distribution of the financing access performance in relation to the influencing factors, we represent the values of the variables DIFF_NB_PROJECTS and DIFF_VAL_PROJECTS on the map (Figures 3 and 4).

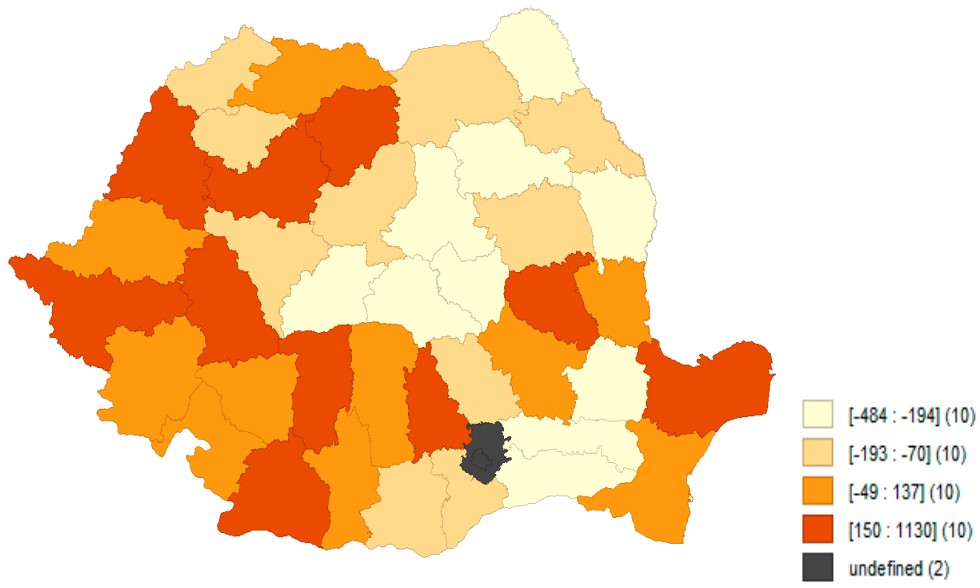

**Figure 3.** Distribution of the variable DIFF_NB_PROJECTS by quartile intervals and by counties. The legend indicates the limits of the quartile intervals for the variable DIFF_NB_PROJECTS and the number of counties in each quartile interval. Source: authors' construction in GeoDa 1.14.

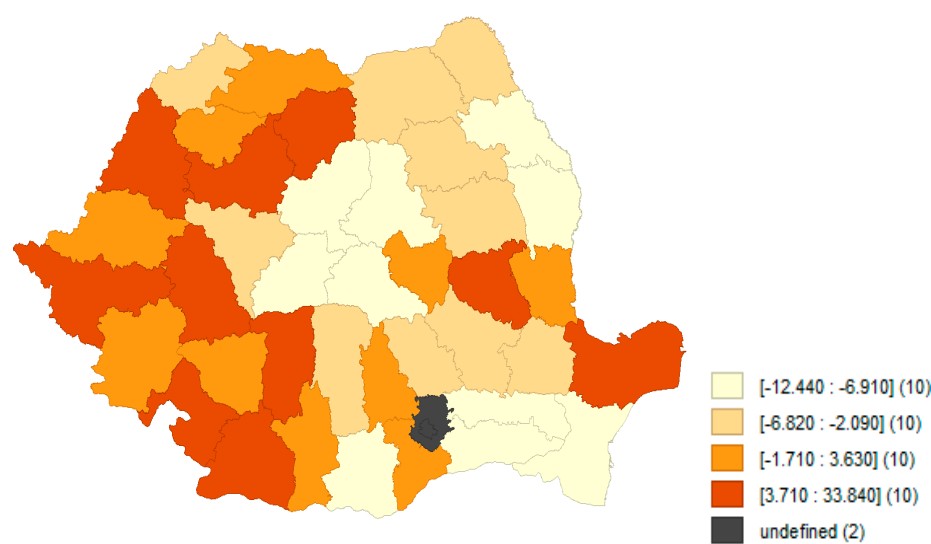

**Figure 4.** Distribution of the variable DIFF_VAL_PROJECTS by quartile intervals and by counties. The legend indicates the limits of the quartile intervals for the variable DIFF_VAL_PROJECTS and the number of counties in each quartile interval. Source: authors' construction in GeoDa 1.14.

For DIFF_NB_PROJECTS, Figure 3 does not indicate clear spatial clusterization, whereas Figure 4 shows important territorial discrepancies between the eastern and the western parts of the country, based on DIFF_VAL_PROJECTS. The western regions are characterized by much higher financed values, conditioned by the explanatory variables in the regressions. Among the latter, both natural factors (agricultural land abundancy and land concentration degree) and motivational ones (human development) are already included. This result is mostly explainable by the capacity of the national institutions administering the projects' application process. In addition, these seem more efficient in the western part of Romania.

Although there is no academic literature on this issue, it would be interesting to look at the average value of submitted projects (Figure 5). Keeping the explanatory variables from the previous regressions, we evaluate by transversal regression models the effect on the variable PROJECT_VALUE.

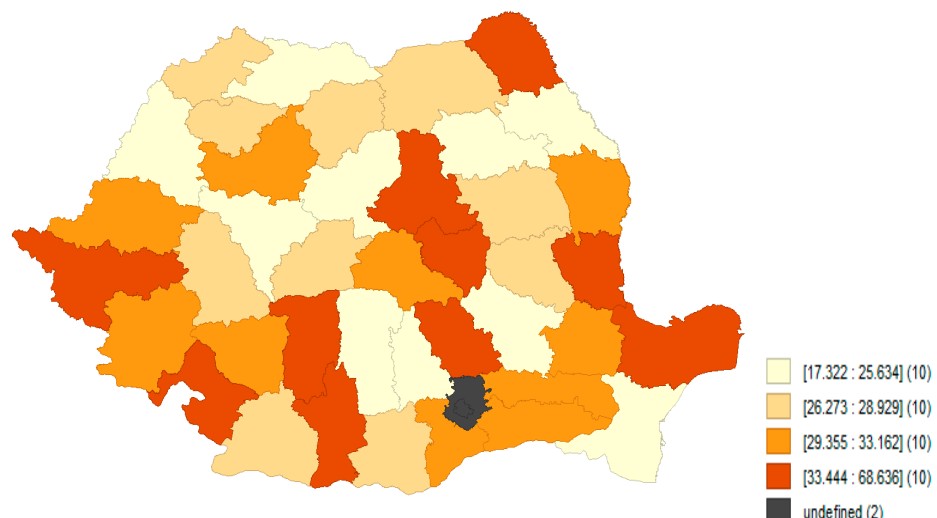

**Figure 5.** Distribution of the variable PROJECT_VALUE by quartile intervals and by counties. The legend indicates the limits of the quartile intervals for the variable PROJECT_VALUE and the number of counties in each quartile interval. Source: authors' construction in GeoDa 1.14.

The regression results on the PROJECT_VALUE variable (Table 7) are inconclusive for almost all explanatory variables. The only significant factor is LHDI, but with the opposite sign compared to the other regressions. The positive sign indicates a preponderance of high value financing in counties with a higher level of human development. This shows that where there are higher human and economic capabilities, it is preferable to develop larger projects at the expense of a multitude of small projects. Higher skilled human resources seem better able to manage large fundable projects.

**Table 7.** Results of OLS regressions on the dependent variable PROJECT_VALUE (coefficients and *t*-stat).

|  | Model 12 | Model 13 |
| --- | --- | --- |
| AREA_AGRIC | $5.76 \times 10^{-6}$ (−0.44) |  |
| RATIO_AGRIC |  | 0.466 (0.54) |
| DENSITY | −0.082 (−1.23) | −0.067 (−0.96) |
| WAGE | −0.0029 (−0.38) | −0.0052 (−0.66) |
| LHDI | ** 38.14 (2.11) | ** 42.23 (2.31) |
| NORTH_WEST | −1.692 (−0.37) | −1.644 (−0.36) |
| CENTER | 1.453 (−0.28) | −1.756 (−0.34) |
| WEST | 3.174 (0.55) | 1.611 (0.26) |
| SOUTH_WEST | * 9.465 (1.91) | * 9.713 (1.96) |
| SOUTH | 2.337 (0.57) | 1.814 (0.43) |
| SOUTH_EAST | 4.792 (1.02) | 2.969 (0.53) |
| NORTH_EAST | reference | reference |
| constant | 18.40 (1.23) | 16.07 (1.09) |
| $R^2$ | 0.366 | 0.368 |

*, ** statistically significant at 10% and 5% level. Source: authors' calculation in STATA 14.

The research hypotheses were partially validated. H1 is valid, but the impact of the agricultural land mostly influences the total value of the projects. The second hypothesis, H2, has a much clearer validation. The land concentration level, computed as the ratio between the available land and the number of exploitations, is highly significant in the regressions. On the contrary, neither the population density nor the average salary level are significant. We validate H3, as lower human development at the regional level is a powerful incentive for applying for funds from the European Union due to less own material and financial resources. At the Member State level, there is positive discrimination through EU policy to support economic convergence. Instead, within each country, the distribution of funds is undertaken by national institutions. Our result shows that, in Romania, regional distribution is similar in the territorial advantage of the poorest, but not through negotiation or administrative regulation but through economic market mechanisms. Specifically, poorer counties and regions are making greater efforts to attract funds because they need them more. H4 is also accepted, there are contagion and diffusion processes. Applying for funding through projects may be associated to a financial behavior for which such mechanisms were already pointed out by the field's literature.

Results of this research cannot be directly compared to the existing literature. Compared to the few empirical studies that deal with the issue of regional or local distribution of European funds for rural development [33–36,41], there are differences in both the econometric methodology and the economic, social, or administrative variables used. However, some similarities can be highlighted. For example, econometric estimates [34] made in Europe's regions (NUTS2) show that the selection of regions for financing Objective 1 (for the development of regions that are economically lagging) is negatively and significantly influenced by per capita income. Instead, the funding for Objective 2 (for regions with declining industrial and rural sectors) is negatively affected by the unemployment rate. However, the sign of the unemployment rate coefficients reveals a perverse effect, contrary to what is desired by European funding: the higher the unemployment rate in a region, the less likely it is to receive funding. One possible explanation may be that

co-financing from the regions is needed. In general, regions with high unemployment rates may be less able to co-finance. Such seemingly contradictory results also emerge from our study. Human development is negatively correlated with the number of applications for funding, but positively correlated with the average value of applications. This indicates that a more deficient human resource is an incentive to make many applications, but less capabilities to manage large projects, which in addition must be co-financed. A study [35] of the spatial distribution of rural development financing between regions and localities in Poland indicates associations with the level of socio-economic development. A concentration of funding is identified around the regional capitals, in localities with a very good situation of local budgets, a high population density, and an advantageous demographic structure. The good state of the demographic indicators is indicated first by a relatively young population and a higher percentage of farmers with higher education. Results also suggest a clustering relationship on the West–East direction but evaluated only descriptively. In the case of Romania, in our study, the spatial econometrics results also indicated phenomena of diffusion and spatial contagion in terms of the number and value of funding applications. An empirical study [36] of the regional distribution of funds in Slovakia identifies spatial interdependencies captured by spatially lagged variables. The direct and indirect effects of the model show that the allocation of funds for rural development is significantly influenced by per capita income and administrative capacity measured as technical efficiency in the production of local public goods. As in the case of Poland, clustering relations have been identified, with major differences in the allocation of funds between the western and eastern regions. In our study on Romania, these effects were also identified by similar spatial econometric methodologies.

## 4. Conclusions

Our study aimed at identifying some determinants of the geographical distribution of rural development funded projects. The practical example chosen was the one of Romania's counties. Results show there are two economic components impacting European funded projects accession. The first deals with the natural but also administrative conditions of the agricultural land. Their abundancy influences the region's performance in respect to intensity of the sums attracted through European projects. Additionally, the land concentration level, given by large farms, is a powerful factor. We can interpret that this component is related to the ability of a region to apply for development funds. The second component is mostly related to the necessity of the funds. Regions and counties with a lower human development access more funds (relatively speaking, conditioned by other factors). The necessity is given by a lower local and regional economic development that implies reduced own material, financial, but mostly human resources.

The spatial analyses conducted emphasized contagion and diffusion territorial processes for European funds access. Neighboring counties and regions act in a similar manner through behavioral mimetics. These phenomena also lead to a longitudinal clusterization, from East to West. The same is suggested by the assessment of the conditioned distributions. Western Romanian regions, which are closer to the European developed countries from both a territorial and a behavior perspective, have local institutions which are more performant in administering the entire process of rural development funds absorption.

Results of this research have the potential to create a framework for the optimum distribution of funds. Ideally, the first component should not act. A too generous amount of funds in regions with higher capabilities for projects administration deepens regional disparities. Spatial distribution of the funds should focus on the second explanatory component, associated to the financing needs. Such results point out that there may be other factors preventing the proper and efficient use of rural development structural funds, factors belonging to the administrative group. Consequently, future studies should include factors/aspects allowing for the identification of the local governmental agencies that are under or over performant in their role as interface between the European Union and

farmers. Consequently, the need for realistic administrative measures in respect to the concrete objectives of funds absorption may be pointed out.

**Author Contributions:** Conceptualization and literature review, D.M., A.-N.R., A.-M.C., I-A.N. and P.-S.L.; variables, data curation, methodology, results, and discussions, D.M., A.-N.R., A.-M.C., I-A.N., I-A.P., T.-I.L. and P.-S.L.; writing—original draft preparation, D.M. and P.-S.L. All authors have read and agreed to the published version of the manuscript.

**Funding:** This research received no external funds.

**Conflicts of Interest:** The authors declare no conflict of interest.

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
