# Peer review of "Territorial Distribution of EU Funds Allocation for Developments of Rural Romania during 2014–2020"

_sustainability, doi:10.3390/su14010506_

Round 1

Reviewer 1 Report

Dear authors, thank you for this interesting paper. One suggestion I have is to further elaborate on the literature and reasoning that is behind your working hypotheses, in order to make a clearer connection between the first sections of the paper and the empirical study.

Author Response

Dear Reviewer,

        Your comment is welcomed. As a result, we have inserted in the article a paragraph of about one page entitled "Determinants of the regional distribution of European funds for rural development". It contains an overview of empirical studies that highlight several economic, social, and administrative factors of the territorial distribution of European funds. (161-194).

Kind regards,

The authors

Reviewer 2 Report

The article deals with current issues in the practical dimension, less in the scientific dimension. The laboriousness of the conducted research as well as the application of interesting research methods are worthy of attention. On the other hand, the article has numerous weaknesses and is not very reader-friendly in terms of communicativeness.

- The title of the article is unfortunate. The use of the term "analysis" is tautological. Furthermore, the use of the term "grant" may be misleading as it is more (I guess) about agricultural development support funds

- The topic is focused on completely Romania. To make it attractive, it would be added in some contexts an international comparison that not only mirrors but also evaluates the country in a comparative sense

- There is a lack of a clearly formulated research objective

- I also did not see any specified motivation for writing the article

- Section 1 is overly descriptive

- The literature review is moderately weak and could have been broader. The literature review ends on p.no.7. There is no literature reference on the remaining pages. Consequently, there are no references to the results of other studies, which significantly weakens the scientific dimension of the article

- The formulation of as many as 6 research hypotheses reduces their importance to research conclusions. Besides, these hypotheses are rather undiscovered in the sense of scientific added value. Moreover, e.g. in H5 it is possible to refer to Human development, but not to indicators which only illustrate the reality

- In many places e.g. in the section 3 the communication of the content is very confusing. The methodological part could be better explained. It is difficult to see what the authors mean by "project" in the dependent variable NB_PROJECTS (the title says "grant")

- It is difficult to agree with the statement that "A current policy in this regard is to subsidize the prices of agricultural products". Currently, under the EU CAP, support is provided not through prices, but mainly through income using direct payments.

- I do not understand the authors in the sentences: "In the economic literature there are studies that highlight unwanted economic discrepancies caused by funding from European funds. It is also shown that there are major disruptions of convergence processes”. Which unwanted economic discrepancies do you mean? What do major disruptions of convergence processes mean?

- In the context of disparities in regional development, it should be noted that the absence of these funds would create even greater disparities (87-102)

- Has the collinearity of the studied variables been investigated? (tab.1)

- It is not clear how the index of local human development was calculated

- It is difficult to understand from the existing description how the variable Spatial lag of VAL_PROJECTS was created

- What do the legends next to the figures mean, how to interpret it?

- The use of absolute values for variables in most cases results in omitting issues concerning the activity in individual regions in obtaining funds for agricultural development (e.g. number of projects per 1000 farms or per 1000 ha). In the case of absolute values, a greater importance of agriculture in a given region translates, for example, into a higher obtaining of funds for agriculture. The conclusions are then less revealing

- "Maps in figures 1 and 2 show that both dependent variables have a similar spatial positioning...." (261).  This is not a discovery since the latter variable follows from the former. However, it would be interesting to know to what extent these variables differ and why?

- “Agricultural properties” (310) - what does this mean?

- I don’t understand where the equations (367, 368) are from, on which data they are based (from which tables ?)

- "...the impact of the intensive factor, namely available agricultural land..." (413-414)? Is the intensive factor- agricultural land. Why?

- "... lower human development is a powerful incentive for applying for money from the European Union..." (418-419). This does not need to be investigated because it follows that less developed regions are supported more strongly within the EU.

- The first part of Conclusions is wordy

- "...Additionally, the land concentration level, given by large farms, is a powerful factor. We may interpret that this component is linked to the ability of a region to apply sustainable projects...." (441-442). Which sustainable projects are concerned?

Author Response

Dear Reviewer,

We revised the manuscript point by point in according to your referee, as follows:

Q: The title of the article is unfortunate. The use of the term "analysis" is tautological. Furthermore, the use of the term "grant" may be misleading as it is more (I guess) about agricultural development support funds

A:  We changed the title according to the suggestion made. (2-3).

Q:The topic is focused on completely Romania. To make it attractive, it would be added in some contexts an international comparison that not only mirrors but also evaluates the country in a comparative sense

A: We have added some statistics on total EU funding for rural development. Also, we completed for Romania some statistical data, to better visualize the importance of financing the programs in Romania in relation to the total European financing. (153-159; 219-224).

Q:  There is a lack of a clearly formulated research objective. I also did not see any specified motivation for writing the article.

A: We have formulated much more clearly, in a separate paragraph, the objective and motivation of the research. (226-237).

Q:  The literature review is moderately weak and could have been broader. The literature review ends on p.no.7. There is no literature reference on the remaining pages. Consequently, there are no references to the results of other studies, which significantly weakens the scientific dimension of the article

A: All bibliographic references refer to economic theory, some empirical studies and the methodology used. However, we have not identified in the literature studies that evaluate the same causal relationships with those in our research. As a result, the discussions are mostly about their own results. However, we have also inserted some discussions regarding some studies that refer to similar issues. (506-511).

Q: The formulation of as many as 6 research hypotheses reduces their importance to research conclusions. Besides, these hypotheses are rather undiscovered in the sense of scientific added value. Moreover, e.g. in H5 it is possible to refer to Human development, but not to indicators which only illustrate the reality

A: For the H5 hypothesis we specified that human development is a very general and comprehensive concept, but in our study, we will use a proxy variable.We do not consider that 6 research hypotheses reduce the importance of research. The main argument is that several hypotheses can be validated or invalidated simultaneously by the same econometric model, by the statistical significance of the explanatory variables. (268-272).

Q:  In many places e.g. in the section 3 the communication of the content is very confusing. The methodological part could be better explained. It is difficult to see what the authors mean by "project" in the dependent variable NB_PROJECTS (the title says "grant")

A: To clarify the presentation, we have corrected the text everywhere, avoiding any terms that can be considered synonymous: grants / projects / applications. An application for funding from a farmer or an economic entity is now referred to in the article as a project.We have now clarified in the text of the article, to avoid possible misunderstandings of the terms used.

Q:  It is difficult to agree with the statement that "A current policy in this regard is to subsidize the prices of agricultural products". Currently, under the EU CAP, support is provided not through prices, but mainly through income using direct payments.

A: We have corrected the sentence as follows: “A current policy in this regard is to finance agriculture by increasing the incomes of agricultural producers, which thus influences lowering the prices of agricultural products.”

Q: I do not understand the authors in the sentences: "In the economic literature there are studies that highlight unwanted economic discrepancies caused by funding from European funds. It is also shown that there are major disruptions of convergence processes”. Which unwanted economic discrepancies do you mean? What do major disruptions of convergence processes mean?In the context of disparities in regional development, it should be noted that the absence of these funds would create even greater disparities (87-102)

A:  We reformulated the idea from the sentences, showing that in the economic literature there are different opinions regarding the effects of financing from European funds on the processes of economic convergence: “The economic literature does not have unanimous opinions on the effects of European funding on the processes of economic convergence. Some studies consider that inadequately accessed funding can increase the heterogeneity of economic development. […] Instead, other research shows convergence effects because of funding through European Union cooperation mechanisms.” In addition, I have cited several studies that address these effects.

Q:  Has the collinearity of the studied variables been investigated? (tab.1)

A: The multicollinearity problem was approached by calculating the correlations between the variables (Table 4). Explanatory variables that were identified as highly correlated were alternately introduced into regressions. We mentioned this in the text.

Q:  It is not clear how the index of local human development was calculated

A:  The Local Human Development Index (LHDI) in not an index computed by the authors. It is an index that replicates the famous Human Development Index (HDI) constructed by the United Nations (UN), for the Romanian counties and localities. It was developed by Sandu for the World Bank [39], Romania Office. It is an index which is officially accepted in Romania as a means to locally measure the level of human development. All the details related to it may be found in Burduja, S. I., Gaman, F., Giosan, V., Glenday, G., Huddlestan, E. N., Ionescu-Heroiu, M., Ha Ti Tu Vu. Identification of project selection models for the regional operational program 2014–20202014, Washington, DC: World Bank.

We have included all this information in the article.

Q:  It is difficult to understand from the existing description how the variable Spatial lag of VAL_PROJECTS was created

A:  We have added such information in the methodological description:

“The SAR model includes the spatial lag of the dependent variable. This is, actually, the weighted average value of the dependent in the neighbours of each spatial unit (namely county) analyzed. Consequently, it allows for the comparison of the value in each county with the average one in the neighbours, allowing for the emphasis of similar or different behaviours in respect to European funding, in space.”

Q:  What do the legends next to the figures mean, how to interpret it?

A: We corrected in the text, explaining under each graphic how the legend is interpreted.

Q:  The use of absolute values for variables in most cases results in omitting issues concerning the activity in individual regions in obtaining funds for agricultural development (e.g. number of projects per 1000 farms or per 1000 ha). In the case of absolute values, a greater importance of agriculture in a given region translates, for example, into a higher obtaining of funds for agriculture. The conclusions are then less revealing

A: We totally agree with the reviewer's observation, his proposal could have been a serious alternative. The use of relative values ​​(e.g. number of projects per 1000 farms or per 1000 ha) would have eliminated the influence of the agricultural area on the number of projects, leaving only the significance of the other factors to emerge from the regressions. The path we chose, using absolute values, shows the significance of the agricultural area on the dependent variables, which could be obvious even without regression analysis. However, our results may highlight the significance of other factors, under the control of the agricultural area as a variable. However, the path proposed by the reviewer can be quite laborious, as several variables with relative values ​​can be created. We can divide both the number of projects and their value by the number of farms, agricultural area, arable land, rural population, etc., thus operating with too many indicators.

Q:  "Maps in figures 1 and 2 show that both dependent variables have a similar spatial positioning...." (261).  This is not a discovery since the latter variable follows from the former. However, it would be interesting to know to what extent these variables differ and why?

A: Indeed, it was to some extent expected to obtain a similar spatial distribution for the number and value of the projects. To answer the extent to which the two variables differ, we constructed a new variable (PROJECT_VALUE), defined as the ratio of the first two. The behavior of this new variable was analyzed geographically (Figure 5) and by regressions (Table 3).

Q:  “Agricultural properties” (310) - what does this mean?

A: We have corrected the term "agricultural properties", replacing it with the term "farms".

Q:  I don’t understand where the equations (367, 368) are from, on which data they are based (from which tables ?)

A: The equations come from simplified regressions, which include only statistically significant factors. Initially, we did not include the results of these estimates in the article, but we have now added them in Table 6. (430-441).

Q:  "...the impact of the intensive factor, namely available agricultural land..." (413-414)? Is the intensive factor- agricultural land. Why?

A: We have corrected the statement, avoiding the term "intensive factor" which can be controversial.

Q:  "... lower human development is a powerful incentive for applying for money from the European Union..." (418-419). This does not need to be investigated because it follows that less developed regions are supported more strongly within the EU.

A: We explained better in the text the result related to the H5 hypothesis, emphasizing the differences that exist in the mechanisms of distribution of funds between countries versus the mechanisms of distribution within the national territory.

Q:  The first part of Conclusions is wordy

A: We have removed the first part of the conclusions. These are now much more concise and start directly with the synthesis of results and policy implications.

Q:  "...Additionally, the land concentration level, given by large farms, is a powerful factor. We may interpret that this component is linked to the ability of a region to apply sustainable projects...." (441-442). Which sustainable projects are concerned?

A: We corrected the text.

Kind regard,

The authors

Round 2

Reviewer 2 Report

The article has a higher standard than before but still needs improvement. The authors have taken some of my comments too lightly.  

  • The topic is focused on completely Romania. To make it attractive, it would be added in some contexts an international comparison that not only mirrors but also evaluates the country in a comparative sense. This comment has been almost entirely omitted in terms of implementing appropriate changes to the text. As it stands, the article gives the impression of being isolated from processes taking place in other countries or regions. After all, it is possible to refer to similar problems in other countries, so that an international background of the research takes place. Therefore, the study is similar, in some places, to the report
  • The motivation of the research is still insufficiently presented
  • I disagree that the results of this research cannot be compared to the existing literature (720). After all, the point is not that the same research methods were used in other studies. What matters more are general trends, regularities. It is enough to use the cited research results [29-32, 38].
  • H5 should be formulated: Human development indices are negative determinants of accessing funds for rural development. Furthermore, I suggest once again considering the number of hypotheses and abandoning H3 and H4.
  • The dependent variable NB_PROJECTS is still not sufficiently explained
  • "The current policy in this regard is to finance agriculture by increasing the incomes of agricultural producers, which thus influences lowering the prices of agricultural products". The second part of this sentence is questionable in the light of existing price trends (https://www.fao.org/worldfoodsituation/foodpricesindex/en/)
  • "...Our study also allows for identification of the local governmental agencies that are under or over performant in their role as interface between the European Union and farmers..." (458-459). I did not notice that this was realized in the article. The authors omitted this remark

Author Response

Dear reviewer,

We have reviewed the article point by point, taking into account your recommendations:

  • The topic is focused on completely Romania. To make it attractive, it would be added in some contexts an international comparison that not only mirrors but also evaluates the country in a comparative sense. This comment has been almost entirely omitted in terms of implementing appropriate changes to the text. As it stands, the article gives the impression of being isolated from processes taking place in other countries or regions. After all, it is possible to refer to similar problems in other countries, so that an international background of the research takes place. Therefore, the study is similar, in some places, to the report.
  • We have responded to these observations in two paragraphs. First of all, in order to show that the concern regarding the distribution of European funds between the regions of a country is not a concern only in Romania, we reviewed some studies in the literature that approach similar topics for other Eastern European countries (105-138). Secondly, I inserted a paragraph describing Romania's situation regarding the absorbed European funds. We also discussed some descriptive statistics regarding the comparison between Romania and the EU averages in relation to some significant variables of financing rural development (230-266).

  • The motivation of the research is still insufficiently presented
  • We have completely reworked the motivation, which is now much more extensive and detailed (298-351).

  • I disagree that the results of this research cannot be compared to the existing literature (720). After all, the point is not that the same research methods were used in other studies. What matters more are general trends, regularities. It is enough to use the cited research results [29-32, 38].
  • We reworked this part of the results, including comparisons between the results of our study and the results of the cited studies (616-647).

  • H5 should be formulated: Human development indices are negative determinants of accessing funds for rural development. Furthermore, I suggest once again considering the number of hypotheses and abandoning H3 and H4.
  • We reformulated the H5 hypothesis as a result of the recommendations received. We abandoned hypotheses H3 and H4 and renumbered the remaining hypotheses accordingly.

  • The dependent variable NB_PROJECTS is still not sufficiently explained.
  • Because there was not enough space in the table to explain the variables, we introduced a footnote that better explains the terms used.

  • "The current policy in this regard is to finance agriculture by increasing the incomes of agricultural producers, which thus influences lowering the prices of agricultural products". The second part of this sentence is questionable in the light of existing price trends (https://www.fao.org/worldfoodsituation/foodpricesindex/en/)
  • We rephased this sentence, as can be seen in the revised manuscript. We are aware of the fact that food prices are increasing, but this is mainly due to the present Covid-19 pandemic that has put pressure on all sectors of activity. In the specific sentence we are speaking about one of the general goals of agricultural financing that, obviously, it may not come true in practice. 

  • "...Our study also allows for identification of the local governmental agencies that are under or over performant in their role as interface between the European Union and farmers..." (458-459). I did not notice that this was realized in the article. The authors omitted this remark.
  • Indeed, here is a mistake coming from our side. What we actually wanted to say was that our results point out the need to further assess administrative factors that may alter the efficiency of using rural development structural funds. We have rephrased the paragraph in this respect:

„Such results point out that there may be other factors preventing the proper and efficient use of rural development structural funds, factors belonging to the administrative group. Consequently, future studies should include factors/ aspects allowing for the identification of the local governmental agencies that are under or over performant in their role as interface between the European Union and farmers. Consequently, the need for realistic administrative measures in respect to the concrete objectives of funds absorption may be pointed out.”

Best regards,

The authors
